# Assessing the accuracy of spectral indices obtained from Sentinel images using field research to estimate land degradation

**Akhtar Ebrahimi[1], Farhad Zolfaghari[1]\*, Marzieh Ghodsi[2], Fatemeh Narmashiri[1]**

**1** Higher Education Complex of Saravan, Saravan, Iran, **2** Faculty of Geography, University of Tehran, Tehran, Iran

\* Zol.farhad@gmail.com

**Data Availability Statement:** All relevant data are within the paper and its Supporting Information files.

## Abstract

Wind erosion resulting from soil degradation is a significant problem in Iran's Baluchistan region. This study evaluated the accuracy of remote sensing models in assessing degradation severity through field studies. Sentinel-2 Multispectral Imager's (MSI) Level-1C satellite data was used to map Rutak's degradation severity in Saravan. The relationship between surface albedo and spectral indices (NDVI, SAVI, MSAVI, BSI, TGSI) was assessed. Linear regression establishes correlations between the albedo and each index, producing a degradation severity map categorized into five classes based on albedo and spectral indices. Accuracy was tested with 100 ground control points and field observations. The Mann-Whitney U-Test compares remote sensing models with field data. Results showed no significant difference (P > 0.05) between NDVI, SAVI, and MSAVI models with field data, while BSI and TGSI models exhibited significant differences (P ≤ 0.001). The best model, BSI-NDVI, achieves a regression coefficient of 0.86. This study demonstrates the advantage of remote sensing technology for mapping and monitoring degraded areas, providing valuable insights into land degradation assessment in Baluchistan. By accurately identifying severity levels, informed interventions can be implemented to mitigate wind erosion and combat soil degradation in the region.

## 1. Introduction

Land degradation is a significant environmental issue in dry areas, leading to soil productivity loss, vegetation decline, and increased desertification. Studying soil and vegetation changes is crucial for understanding these processes' dynamics and developing effective land management and conservation strategies. Reflectance from the soil and vegetation surface plays a significant role in monitoring environmental changes [1]. Vegetation stabilizes dry areas, reducing erosion effects and determining the intensity of desertification [2]. Reduction in vegetation cover and biomass directly contributes to land degradation, increasing surface albedo [3].

Various factors contribute to land degradation in dry areas, including ecosystem changes, landslides, deforestation, human impact, and soil erosion [4]. In degraded areas with thin

**Funding:** The authors received no specific funding for this work.

**Competing interests:** The authors have declared that no competing interests exist.

vegetation, albedo increases, indicating higher surface reflectance [5]. The decrease in vegetation cover reduces humidity, further increasing surface albedo in these regions [6]. Monitoring changes in albedo and vegetation indices can help identify and assess the extent of land degradation [7].

Remote sensing technology has proven valuable in mapping and monitoring desertification intensity in arid and semi-arid regions. Previous studies have utilized remote sensing data, such as Landsat Operational Land Imager (OLI) satellite imagery, to develop classification algorithms and spectral indices for assessing desertification [8–10]. In 2011, researchers proposed a model based on the albedo index and normalized vegetation difference index, effectively reflecting the desertification of the Earth's surface and changes in desertified areas [3].

Spectral composition analysis has been employed to study vegetation, water, and bare soil using satellite imagery, reducing data noise through techniques like Minimum Noise Fraction (MNF) and Pixel Purity Index (PPI) [6]. The state and degree of desertification can be determined by analyzing the spectral-temporal characteristics of vegetation and albedo obtained from Landsat satellite images [11]. Spatio-temporal patterns of desertification have been explored using indicators such as the Modified Soil-Adjusted Vegetation Index (MSAVI), Fractional Vegetation Cover (FVC), and Temperature–Vegetation Drought Index (TVDI), alongside surface temperature and albedo [12].

The status of the land surface regarding vegetation biomass, landscape pattern, and climatology at a small scale has been investigated and evaluated using the Normalized Difference Vegetation Index (NDVI), Topsoil Grain Size Index (TGSI), and Albedo indices [10]. The researchers did not observe any correlation between the NDVI and Albedo indices or between the NDVI and TGSI. However, they found a significant correlation (0.77 to 0.92) between the Albedo and TGSI indices in non-desertification areas. In this study, they demonstrated that based on these indices, the region's desertification is increasing. They also showed that by identifying changes in the levels of NDVI, TGSI, and surface Albedo, the process of desertification could be identified, and the highest TGSI values were observed in areas experiencing severe desertification.

Researchers have employed the extraction of NDVI, Bare soil index (BSI), and Albedo indices from Landsat satellite images and the use of the Change vector analysis (CVA) model to investigate the determinants of the direction and magnitude of these indices and evaluate the degradation or improvement of land conditions over different periods [13]. This study demonstrated the relationship between NDVI and Albedo as an indicator of desertification. Additionally, other researchers focused on studying soil degradation using the Albedo and Modified soil-adjusted vegetation index (MSAVI) indices, showcasing the intensity of soil degradation through remote sensing technology [14]. The examination and study of a regional geographical model, along with the investigation of three indices (NDVI, TGSI, MSAVI), and surface Albedo revealed that the relationship between the Albedo and MSAVI indices is more suitable for obtaining a more robust and appropriate desertification intensity map compared to the other two indices [15]. The assessment of desertification intensity based on the spatial characteristics of Albedo and NDVI, as well as Albedo and MSAVI, showed that a decrease in vegetation coverage in the studied areas led to a decrease in the values of NDVI and MSAVI while increasing the Albedo values [16, 17].

Various methods have been used in the Sistan and Baluchestan provinces to evaluate and classify desertification intensity. However, challenges such as high costs, regional size, and limited access due to security issues hinder comprehensive assessments [18–22].

The use of multispectral satellite data, such as Sentinel-2, for identifying compacted soil has been shown to be effective [23]. Furthermore, the results of studying two indices, NDVI and BSI, using Sentinel-2 images indicated that, with a 95% confidence level, vegetated areas and

bare soil areas could be distinguished using these two indices [23]. The BSI index has been widely utilized for identifying areas with bare soil and no vegetation cover [24]. The validation of different spectral indices and field studies is crucial for determining the most accurate spectral index to assess land degradation. This validation is essential due to the characteristics of the type of cover and the unique conditions of areas affected by destruction. The present study aims to evaluate the performance of spectral indices obtained from remote sensing and field studies in the Rutak Saravan region to enhance the accuracy of land degradation assessment. The indicators extracted from Sentinel-2 data, such as NDVI, MSAVI, SAVI, BSI, TGSi, and albedo, as well as those investigated in the field, including soil, vegetation, wind erosion, and climate criteria of the region, were evaluated based on Iranian Model of Desertification Potential Assessment (IMDPA). The study aims to identify the most reliable indicator with minimal error by comparing the results with field data. This research will contribute valuable insights for monitoring and managing land degradation in the region, aiding in sustainable land management and environmental conservation efforts.

## 2. Materials and methods

### 2.1 Study area

The study area, encompassing 46,656.5 hectares, is situated within the latitude range of 28° 00′ to 28° 16′ North and the longitude range of 62° 29′ to 62° 47′ East (Fig 1). The Rutak-Saravan region is located approximately 150 km away from the center of Saravan City in Sistan and Baluchistan province, Iran. The elevation in the region ranges from 530 meters in the western part to 480 meters in the eastern part, near the border of Pakistan. Geomorphologically, the region belongs to the Pediments unit, which extends towards the Mashkel playa in Pakistan, as determined by the slope index.

The region experiences a Mediterranean rainfall regime, with a peak winter precipitation of 51.15 mm and an average long-term precipitation of 35.11 mm. The annual average temperature is 22.1°C. The hottest month is July, with an average temperature of 39.7°C, while the coldest month is January, with an average temperature of 16.7°C [22]. The vegetation in the area is predominantly comprised of bush and shrub species, which are sparsely distributed in the flood plains. Due to inadequate precipitation, several areas of the vegetation cover have desiccated. Herbaceous and annual species have nearly vanished, leading the land that was

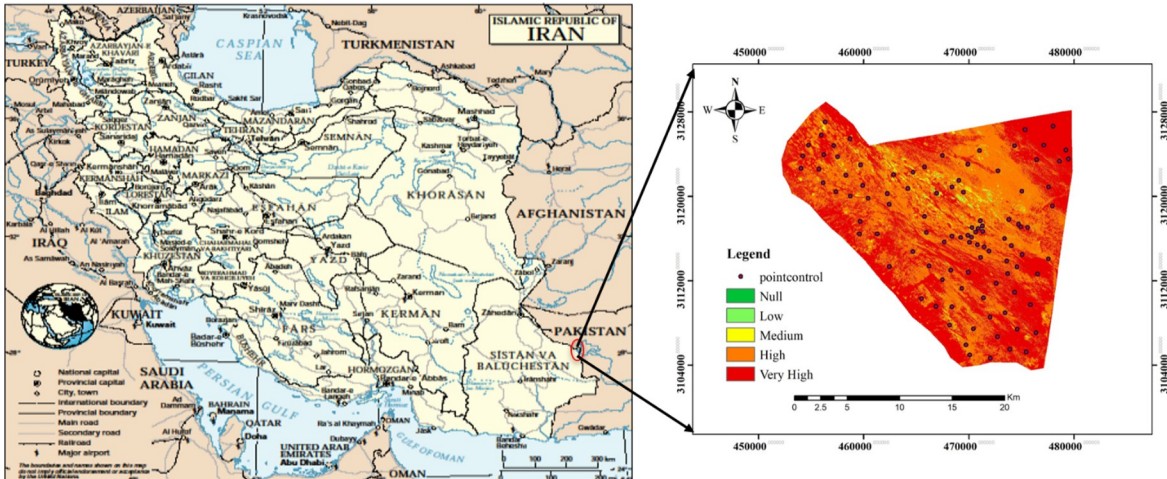

**Fig 1. Location of the study area (http://earthobservatory.nasa.gov/).**

previously utilized as pasture for light livestock to transform into completely arid regions with scanty bush cover.

## 2.2 Softwares

The Sentinel Application Platform (SNAP) software was used to preprocess and process Sentinel satellite images. Additionally, statistical analysis and regression were conducted using $SPSS_{25}$ software. The degradation severity map was generated as an output using $ArcGIS_{10.6}$ software. These software tools were chosen for their established reliability and functionality in remote sensing data analysis and geographic information system (GIS) mapping. Using SNAP allowed for efficient handling and manipulation of the Sentinel satellite images. $SPSS_{25}$ facilitated the statistical analysis and regression modeling necessary for evaluating the relationship between spectral indices and albedo. $ArcGIS_{10.6}$, on the other hand, provided a robust platform for spatial analysis and the creation of the degradation severity map. The study aimed to ensure accurate and reliable results in assessing land degradation in the Rutak-Saravan region by employing these software applications.

## 2.3 Remote sensing dataset and preprocessing

The initial phase involved the preparation of satellite images and the execution of image processing operations. Sentinel-2A data captured by the MSI-L1C sensor on August 2, 2020, was selected to accomplish this. The required Sentinel data was obtained from the Open Access Hub of the European Space Agency [25].

According to the European Space Agency (ESA) in 2015, the Sentinel-2 Multispectral Imager (MSI) has 13 spectral bands. Four of these bands are in the visible and near-infrared (VIS-NIR) range and have a spatial resolution of 10 meters. Six bands are in the red-edge and short-wave infrared (SWIR) range and have a spatial resolution of 20 meters. The remaining three bands are Coastal aerosol, Water Vapor, and Cirrus, with a spatial resolution of 60 meters. You can find the specifications of these spectral bands is available at sentinel-2 website [26, 27].

Subsequently, the SNAP software was utilized to preprocess and process the Sentinel satellite images. Table 1 displays the soil and Vegetation indices and their equations that were utilized to detect degradation intensity.

The wavelengths of the bands used in the equations are translated to the central bands of Sentinel-2. In this equation, L represents the canopy background adjustment factor, which is equal to 0.5. The values of α and b are 0.96916 and 0.84726, respectively.

**Table 1. The vegetation and soil indices selected for the current study, their respective equation, and bibliographic references.**

| Indices | Algorithm | Goal | References |
|---------|-----------|------|------------|
| NDVI | $\frac{NIR-R}{NIR+R}$ | The evaluation of chlorophyll activity in plants and monitoring the state of vegetation cover are among the primary applications of this technique. | [28, 29] (Rouse et al, 1973; Gadal et al, 2021) |
| MSAVI | $\frac{NIR-R*(1+L)^2}{NIR+R+L}$ | This technique is sensitive to changes in vegetation amount/cover, which can be used to correct soil surface brightness. It is also sensitive to differences in atmospheric conditions between areas or times. | [30] (Albalawi & Kumar, 2013) |
| SAVI | $\frac{(1+L)\times(NIR-R)}{NIR+R+L}$ | The SAVI minimizes the soil influence on vegetation quantification by introducing a soil adjustment factor. It is computed in a range between –1 and 1 | [24, 31] (Mróz & Sobieraj, 2004; Polykretis et al, 2020) |
| TGSI | $\frac{(R-B)}{(R+B+G)}$ | The topsoil grain size index is an index that characterizes the texture of the soil surface based on the soil reflectance curve. | [32] (Xiao et al, 2006) |
| BSI | $\frac{(SWIR+R)-(NIR+B)}{(SWIR+R)+(NIR+B)} \times 100 + 100$ | The Bare Soil Index (BSI) is a spectral index that enhances the detection of exposed soil surfaces and uncultivated areas by relying on soil characteristics. | [23, 33, 34] (Rikimaru et al, 2002; Mzid et al, 2021; Nguyen et al, 2021) |

## 2.4 Surface albedo

The surface albedo index was calculated using Liang's relation, tailored explicitly for Sentinel-2 sensor images. This relation, described by Naegeli et al. (2017), was employed to determine the albedo values according to the Eq (1) [35].

$$\alpha = 0 \cdot 356b_2 + 0 \cdot 130b_4 + 0 \cdot 373b_8 + 0 \cdot 085b_{11} + 0 \cdot 072b_{12} - 0 \cdot 0018 \qquad (1)$$

In this equation, α represents the surface albedo value, and the symbol refers to the band number of the Sentinel-2 optical satellite.

A crucial step was to convert the bands with resolutions of 10 and 20 meters to a consistent spatial resolution of 10 meters to calculate the albedo. This conversion was achieved by utilizing the resampling command in the SNAP software. Accurate and consistent albedo values could be derived by harmonizing the resolution across all bands for further analysis and interpretation.

The investigation focused on the August summer season to minimize errors resulting from surface moisture and the absence of annual species. This period was chosen because the soil surface is typically dry, and only perennial species are present, which play a crucial role in soil stability and protection.

## 2.5 Land degradation severity map

To establish the correlation between vegetation indices (NDVI, SAVI, TGSI, MSAVI, BSI) and Surface Albedo, we employed a linear regression model using 520 randomly selected pixels. We calculate the slope coefficient of the regression line for each vegetation index and albedo to generate the Land degradation severity map by Eq (2).

$$DI = a*Index - Albedo \qquad (2)$$

In this equation, "DI" stands for degradation intensity, and "a" stands for the coefficient value resulting from the regression between the albedo index and the corresponding spectral index (The (a) value is the slope of the orthogonal lines found in the NDVI, SAVI, MSAVI, BSI, and TGSI separately with Albedo relationship), which is determined by dividing one by the coefficient of the vegetation index [13] and "Index" is the examined spectral indices.

The Jenks natural breaks classification method (Jenks, 1967) was employed to classify the data into five degrees of land degradation, namely Null, Low, Medium, High, and Very High [12, 13, 36]. To assess the accuracy of the models based on spectral indices, 100 plots measuring 10 x 10 meters were randomly selected across different parts of the study area, aligning with the pixel size of the Sentinel-2 images. These plots represented varying degrees of land degradation and were sampled during the same season as the time of the Sentinel image, specifically August 2020.

The IMDPA model was employed to quantify the severity of degradation in the study plots. Field studies were conducted to ascertain the degradation intensity in each plot, with the analysis guided by the criteria and sub-criteria derived from the IMDPA procedures. These criteria and sub-criteria were selected based on their relevance to the specific conditions of the study area [22, 37].

The process adopted a scientific methodology to identify the factors contributing to land degradation and to quantify their intensity. The IMDPA model uses four main criteria for this assessment: soil, vegetation cover, wind erosion, and climate. Each criterion is evaluated using selected indices, which are scored on a scale of 1 to 2, reflecting the weight of each factor.

The value for each criterion is then computed as the geometric mean of the scores of the individual indices Eq (3).

$$\text{Index(X)} = [\text{Layer}_1 \times \text{Layer}_2 \times \cdots \times \text{Layer}_3]^{\frac{1}{n}} \tag{3}$$

where:

- Index(X) represents the given criteria,
- Layer represents the index of each criterion, and
- n is the number of indices for each criterion.

The overall degradation intensity is then determined as the geometric mean of the four criteria: soil, wind erosion, climate, and vegetation cover Eq (4).

$$\text{Degradation intensity} = [\text{soil} \times \text{wind erosion} \times \text{climate} \times \text{vegetation cover}]^{\frac{1}{4}} \tag{4}$$

Finally, based on the calculated degradation intensity, each plot was classified into one of four risk categories (S1 Appendix).

## 2.6 Correlation analysis

The Mann-Whitney U-Test was employed to analyze the relationship between each remote sensing model and field observations. Additionally, the Root Mean Square Error (RMSE) was used as a quantitative index to evaluate the performance of the models using both remote sensing and field data. The RMSE indicates the magnitude of error in the models, with a lower value indicating better performance. The calculation of RMSE is based on the Eq (5):

$$\text{RMSE} = \sqrt{\frac{\sum_{K=0}^{K} (X_k - Y_k)^2}{K}} \tag{5}$$

$X_{K:}$ observed data in field research
$Y_{k:}$ values obtained from the model
K: number of data

# 3. Results

## 3.1 Relationship between albedo and spectral indices

The relationship between the NDVI and Albedo indices was investigated through linear regression analysis, revealing a significant negative correlation between these two indices (Fig 2A). The correlation coefficient was determined to be 0.825 at a 99% confidence level. It was observed that as the NDVI index increased, the surface albedo exhibited a decrease.

The results of the linear regression analysis between the SAVI and Albedo indices demonstrated a strong negative correlation with a correlation coefficient of 0.799 at a 99% confidence level (Fig 2B). Similarly, the MSAVI and Albedo indices exhibited a negative correlation with a correlation coefficient of 0.790 (99% confidence level) (Fig 2C). Furthermore, the linear regression analysis between the BSI and Albedo indices revealed a positive correlation coefficient of 0.763 (99% confidence level) (Fig 2D), indicating that an increase in the BSI index corresponds to an increase in the Albedo index.

The relationship between the TGSI and Albedo indices was positively correlated, with a correlation coefficient of 0.836 (99% confidence level) (Fig 2E). The linear regression analysis between the BSI and NDVI indices showed a strong negative correlation with a correlation

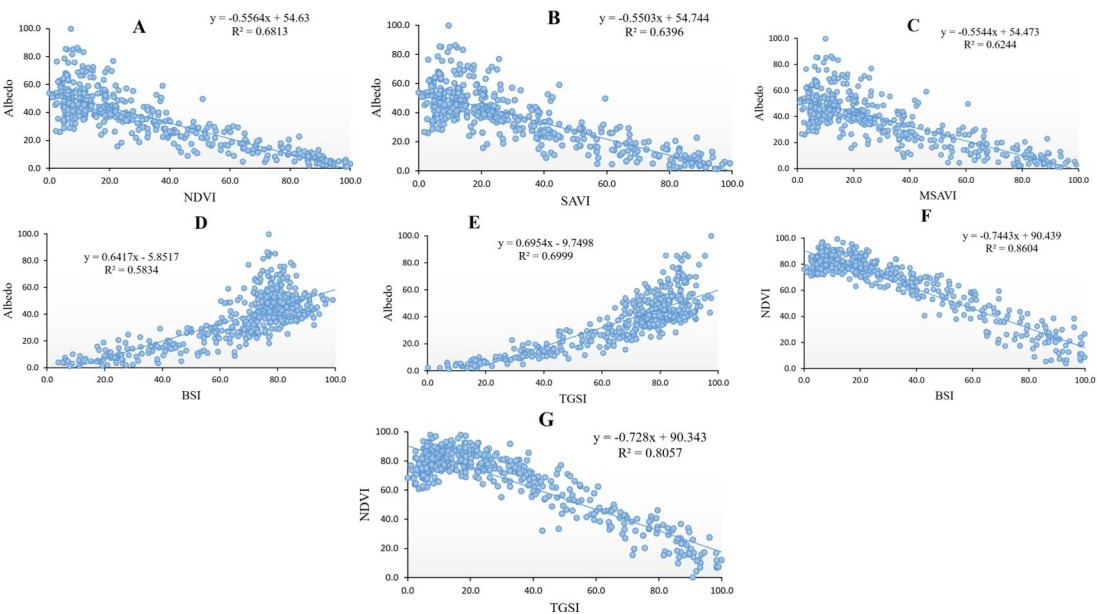

**Fig 2.** Linear regression relationship between albedo and other indicators; A: Albedo-NDVI, B: Albedo-SAVI, C; Albedo-MSAVI, D: Albedo-BSI, E: Albedo-TGSI, F: NDVI-BSI, and G: NDVI-TGSI.

coefficient of -0.927 (99% confidence level) (Fig 2F). Similarly, the TGSI and NDVI indices exhibited a negative correlation with a correlation coefficient of -0.897 (99% confidence level) (Fig 2G).

Overall, the examination of the relations between the Albedo index and each of the investigated vegetation indices (NDVI, SAVI, MSAVI, BSI, and TGSI) revealed that the highest correlation was observed between the TGSI and Albedo indices with a correlation coefficient of +0.836. Conversely, the lowest correlation was observed between the BSI index and albedo, with a correlation coefficient of +0.763 in the studied area.

Furthermore, the analysis of the relationship between NDVI, BSI, and TGSI indicated that the strongest negative correlation existed between NDVI and BSI, with a correlation coefficient of 0.927. Table 3 displays the relationship between surface albedo and each studied index and the correlation between NDVI and BSI/TGSI indices in the studied area.

## 3.2. Land degradation severity map

The severity of land degradation was assessed using the coefficients derived from the linear regression analysis between the Albedo index and the vegetation indices, as presented in Table 2. These coefficients were then utilized to generate the land degradation severity equation, as illustrated in Table 3.

The results of the land degradation severity classification based on the Albedo-NDVI model revealed that the majority of the studied area, comprising 44.32%, falls into the very high degradation class. In contrast, less than one percent (0.36) of the area is classified as having no degradation (Fig 3A). Furthermore, classification based on the Albedo-SAVI model demonstrated that the highest level of degradation (43.11% of the area) was observed in the very high degradation class, while the class representing areas without degradation accounted for only 0.45% of the region (Fig 3B).

Analyzing the severity of degradation using the Albedo-MSAVI model demonstrated that the highest level of degradation, at 41.78%, was observed in the high class. In contrast, the

**Table 2. The results of the linear regression relationship between the albedo and spectral indices.**

| Variables | Regression equation | Regression coefficient | Std. error | P value |
|---|---|---|---|---|
| NDVI-Albedo | Albedo = -0.5564 × NDVI + 54.63 | 0.6813 | -0.825 | 0.0001 |
| SAVI-Albedo | Albedo = -0.5503 × SAVI + 54.744 | 0.6396 | -0.799 | 0.0001 |
| MSAVI-Albedo | Albedo = -0.5544 × MSAVI + 54.473 | 0.6244 | -0.790 | 0.0001 |
| BSI-Albedo | Albedo = +0.6417 × BSI—5.8517 | 0.5834 | +0.763 | 0.0001 |
| TGSI-Albedo | Albedo = +0.6954 × TGSI—9.7498 | 0.6999 | +0.836 | 0.0001 |
| BSI-NDVI | NDVI = -0.7443 × BSI + 90.439 | 0.8604 | -0.927 | 0.0001 |
| TGSI-NDVI | NDVI = -0.728 × TGSI + 90.343 | 0.8057 | -0.897 | 0.0001 |

lowest level, at 0.46%, was found in the without degradation class (Fig 3C). Similarly, the Albedo-BSI model indicated that the highest level of degradation, with 37.85%, occurred in the high class. In contrast, the lowest level, at 0.56%, was observed in the without degradation class (Fig 3D).

Regarding the Albedo-TGSI model, the analysis revealed that the very high class represented the medium level of degradation, covering 32.78% of the area. In contrast, the null degradation class accounted for the lowest level, encompassing only 5.38% of the region (Fig 3E).

According to the NDVI-BSI model, the severity of degradation in the area was observed in the high degradation class, which covers 37.81% of the total area. In contrast, the null degradation class, which represents the lowest level of degradation, only covers 0.58% of the area (Fig 3F). Meanwhile, the NDVI-TGSI model revealed that the very high degradation class had the highest level of degradation in the area, covering 48.01% of the total area. Conversely, the null degradation class accounted for only 0.25% of the total area, representing the lowest level of degradation (Fig 3G). The percentage of land degradation based on each model is summarized separately in Table 4.

### 3.3 Mann-Whitney test results

The Mann-Whitney test was conducted to assess the severity of land degradation using the Albedo index and various spectral indices derived from remote sensing and field data (Table 5). The results revealed that in the studied area, the severity of degradation based on the NDVI, SAVI, and MSAVI indices did not exhibit a significant difference compared to the field data at a 95% confidence level ($\alpha = 0.05$), with p-values greater than 0.05. However, the BSI and TGSI indices showed a significant difference with the field data, as indicated by p-values of 0.00. Furthermore, the Mann-Whitney test demonstrated that the BSI-NDVI model did not significantly differ from the field data (p-value = 0.006). In contrast, the TGSI-NDVI model showed a significant difference with a p-value of 0.336.

**Table 3. Land degradation severity equation.**

| Indices | Equation |
|---|---|
| NDVI-Albedo | I = 1.797∗NDVI − Albedo |
| SAVI-Albedo | I = 1.817∗SAVI − Albedo |
| MSAVI-Albedo | I = 1.803∗MSAVI − Albedo |
| BSI-Albedo | I = 1.558∗BSI + Albedo |
| TGSI-Albedo | I = 1.438∗TGSI + Albedo |
| BSI-NDVI | I = 1.343∗BSI − NDVI |
| TGSI-NDVI | I = 1.373∗TGSI − NDVI |

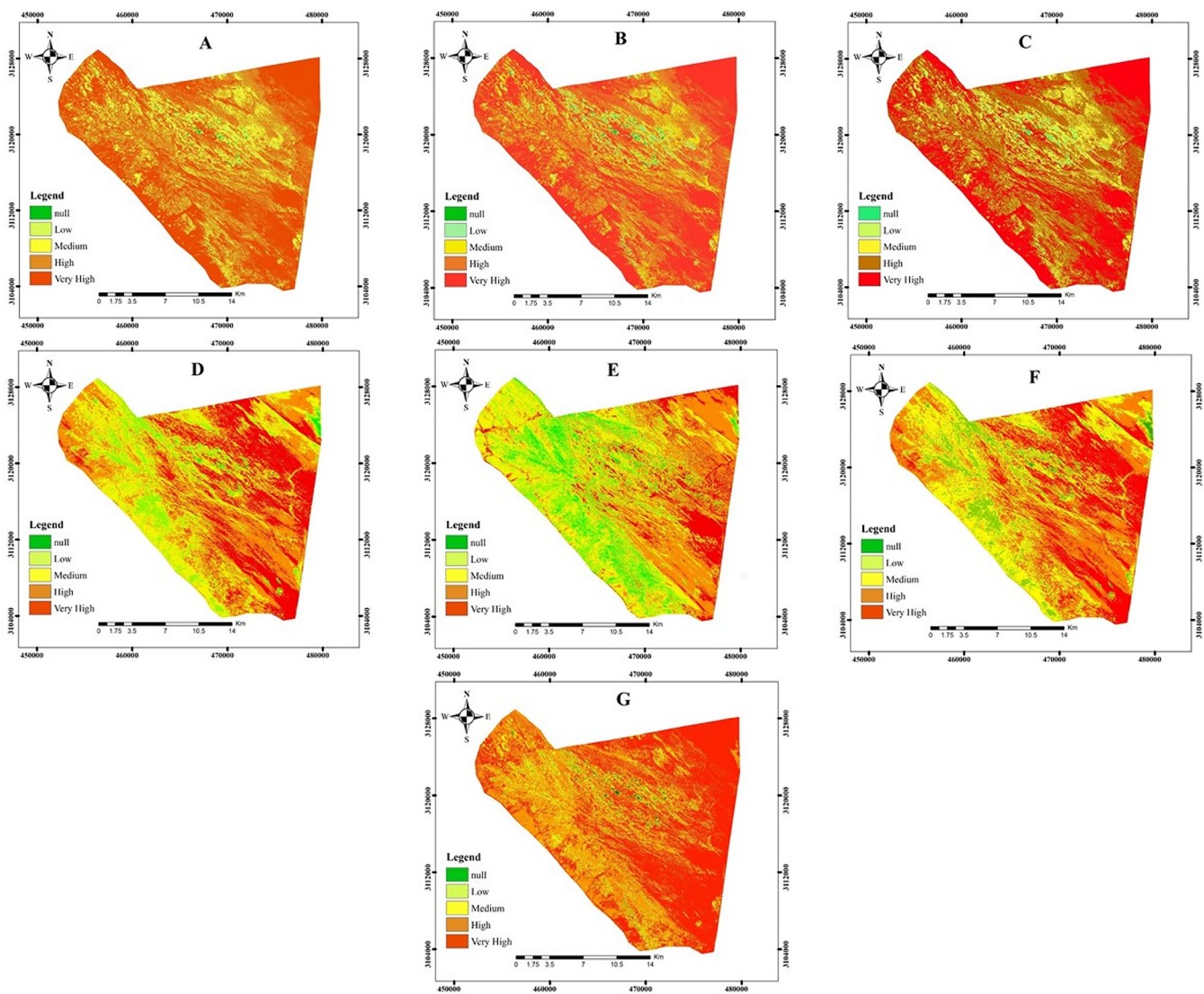

**Fig 3. Land degradation severity map based on albedo index and spectral indices.** A: Albedo-NDVI, B: Albedo-SAVI, C; Albedo-MSAVI, D: Albedo-BSI, and E: Albedo-TGSI, F: NDVI-BSI, and G: NDVI-TGSI (*Contains modified Copernicus Sentinel data [MSI-L1C sensor on August 2, 2020]*).

## 3.4 Model evaluation and comparison with field data based on RMSE

The comparison of RMSE values between the Albedo models, spectral indices, and field-observed data revealed that in the studied area, the MSAVI-Albedo indices exhibited the highest consistency with the field survey data (RMSE = 0.65), as shown in Table 6. An RMSE value of zero or lower indicates a perfect fit for the data. The results of linear regression analyses between the Normalized Difference Vegetation Index (NDVI) and the Topsoil Grain Size Index (TGSI) (0.89) and between NDVI and the Bare Soil Index (BSI) indicate (0.92) a high correlation coefficient between NDVI and these indices (Fig 2F and 2G), with a negative relationship between these indices. The behavior of TGSI and BSI with NDVI was similar to that of the Albedo index. This suggests that TGSI and BSI indices can also be used in preparing the degradation intensity map. The study compared and tested the accuracy of the model prepared with the albedo index, vegetation indices, TGSI, and BSI indices with the NDVI index using

**Table 4. Percentage of land degradation severity classes based on the (I = a\*Index-Albedo) model.**

| Severity degradation Class | | IMDPA | NDVI-TGSI | NDVI-BSI | TGSI |
|---|---|---|---|---|---|
| Very high | Range | (1.75)-(2.00) | (0.071)-(0.255) | (149.88)-(159.81) | (0.302)-(0.444) |
| | Area percentage | 30.00 | 48.01 | 21.32 | 11.40 |
| High | Range | (1.50)-(1.74) | (0.025)-(0.071) | (147.44)-(149.88) | (0.244)-(0.302) |
| | Area percentage | 40.00 | 39.21 | 37.81 | 31.45 |
| Medium | Range | (1.25)-(1.49) | (-0.045)-(0.025) | (145.01)-(147.44) | (0.169)-(0.244) |
| | Area percentage | 19.00 | 10.11 | 30.02 | 32.78 |
| Low | Range | (1.00)-(1.24) | (-0.205)-(-0.045) | (139.33)-(145.01) | (-0.016)-(0.169) |
| | Area percentage | 6.00 | 2.43 | 10.26 | 18.99 |
| Null | Range | (1.00)> | (-0.741)-(-0.205) | (108.11)-(139.33) | (-0.42)-(-0.016) |
| | Area percentage | 5.00 | 0.25 | 0.58 | 5.38 |
| **Severity degradation Class** | | BSI | MSAVI | SAVI | NDVI |
| Very high | Range | (174.11)-(185.62) | (0.041)-(0.727) | (0.060)-(0.791) | (0.097)-(1.00) |
| | Area percentage | 23.06 | 39.72 | 43.11 | 44.32 |
| High | Range | (171.29)-(174.11) | (-0.054)-(0.041) | (-0.044)-(0.060) | (-0.041)-(0.097) |
| | Area percentage | 37.85 | 41.78 | 38.38 | 38.78 |
| Medium | Range | (168.47)-(171.29) | (-0.098)-(-0.054) | (-0.092)-(-0.044) | (-0.104)-(-0.041) |
| | Area percentage | 28.97 | 13.93 | 14.00 | 13.13 |
| Low | Range | (161.89)-(168.47) | (-0.122)-(-0.098) | (-0.119)-(-0.092) | (-0.224)-(-0.104) |
| | Area percentage | 9.55 | 4.11 | 4.06 | 3.42 |
| Null | Range | (125.72)-(161.89) | (-0.295)-(-0.122) | (-0.325)-(-0.119) | (-0.546)-(-0.224) |
| | Area percentage | 0.56 | 0.46 | 0.45 | 0.36 |

**Table 5. Mann-Whitney test results between the model obtained from spectral indices and field data.**

| Index | group | N | Mean Rank | Sum of Ranks | Mann-Whitney U | Wilcoxon W | Z | Asymp. Sig (2-tailed) |
|---|---|---|---|---|---|---|---|---|
| NDVI-Albedo | 1 | 100 | 107.40 | 10739.50 | 4310.500 | 9360.500 | -1.777 | 0.076 |
| | 2 | 100 | 93.61 | 9360.50 | | | | |
| SAVI-Albedo | 1 | 100 | 106.70 | 10669.50 | 4380.500 | 9430.500 | -1.597 | 0.110 |
| | 2 | 100 | 94.31 | 9430.50 | | | | |
| MSAVI-Albedo | 1 | 100 | 105.65 | 10564.50 | 4485.500 | 9535.500 | -1.326 | 0.185 |
| | 2 | 100 | 95.36 | 9535.50 | | | | |
| BSI-Albedo | 1 | 100 | 72.17 | 7217.00 | 2167.000 | 7217.000 | -7.096 | 0.000 |
| | 2 | 100 | 128.83 | 12883.00 | | | | |
| TGSI-Albedo | 1 | 100 | 80.55 | 8054.50 | 3004.500 | 8054.500 | -5.017 | 0.000 |
| | 2 | 100 | 120.46 | 12045.50 | | | | |
| BSI-NDVI | 1 | 100 | 89.81 | 8980.50 | 3930.500 | 8980.500 | -2.742 | 0.006 |
| | 2 | 100 | 111.20 | 11119.50 | | | | |
| TGSI-NDVI | 1 | 100 | 104.21 | 10420.50 | 4629.500 | 9679.500 | -0.962 | 0.336 |
| | 2 | 100 | 96.80 | 9679.50 | | | | |

**Table 6. Quantitative RMSE index comparison between spectral index models and field data.**

| | NDVI-Albedo | SAVI-Albedo | MSAVI-Albedo | BSI-Albedo | TGSI-Albedo | BSI-NDVI | TGSI-NDVI |
|---|---|---|---|---|---|---|---|
| **RMSE** | 0.692 | 0.678 | 0.655 | 2.204 | 2.184 | 1.153 | 0.663 |

field data. The results show that the model prepared with the TGSI index and NDVI has a high accuracy in evaluating the severity of degradation with an RMSE of 0.663. The study suggests that the model prepared based on the NDVI and the TGSI has a high accuracy for identifying the severity of degradation. This model was found to be more accurate after the model was prepared based on the MSAVI-Albedo index.

## 4. Discussion

The objective of this study was to identify the best model for estimating the intensity of land degradation by comparing remote sensing indices using Sentinel satellite data and field data.

The Sentinel satellite images with a 10-meter resolution were used as the basis for creating vegetation and soil cover indices. Seven different combinations of these indices (Albedo-NDVI, Albedo-SAVI, Albedo-MSAVI, Albedo-BSI, Albedo-TGSI, NDVI-BSI, and NDVI-TGSI) were evaluated to assess their impact on identifying degraded areas. The goal was to determine the best combination based on the accuracy of the results obtained through field surveys.

The results of the investigation into the relationship between the Albedo index and vegetation cover indices (NDVI, SAVI, MSAVI, BSI, and TGSI) demonstrated that the highest correlation in the study area was observed between the TGSI-Albedo index, with a value of +0.836. In contrast, the lowest correlation was found between the BSI-Albedo index, with a value of +0.763.

Furthermore, the analysis of the relationship between the NDVI index and the BSI and TGSI indices indicated that the strongest negative correlation existed between the NDVI and BSI indices, with a correlation coefficient of +0.927.

The results of the Mann-Whitney U test examining the intensity of land degradation based on the Albedo index, spectral indices, and field data indicated that in the study area, with 95% confidence ($\alpha = 0.05$), the models of land degradation intensity derived from the NDVI, SAVI, and MSAVI indices did not show a significant difference compared to the models derived from field measurements.

Researchers conducted a study in a part of Mongolia, China, investigating the relationship between three indices, NDVI, TGSI, MSAVI, and Albedo, to generate a desertification intensity map [38]. The results of their study differed from the findings of the present research. They demonstrated that the relationship between the Albedo-MSAVI indices compared to the NDVI and TGSI indices with albedo is more robust and more suitable for determining the desertification intensity map. This difference might be attributed to using satellite images with higher spatial accuracy (Sentinel-2 images with a 10-meter resolution) compared to Landsat images with a 30-meter resolution.

It is a scientific fact that the amount of reflected light from the Earth's surface in the range of 0.2 to 0.6 micrometers increases due to a decrease in vegetation cover [39].

The results of the linear regression analysis between the Albedo index and spectral indices (BSI, MSAVI, SAVI, NDVI, and TGSI) in the Rutak region of Sistan and Baluchestan province indicate a negative relationship (correlation coefficient Albedo-NDVI = -0.82), which is consistent with the findings of other researchers in dry areas [6].

Studies using the Change Vector Analysis (CVA) technique have confirmed using the NDVI and Albedo indices to determine degradation intensity in different periods [11].

Other studies based on the albedo and vegetation cover indices [13] have also shown a strong correlation between the NDVI and Albedo indices, consistent with the present research findings.

Examining various vegetation cover indices and albedo results indicates that an increase in the MSAVI, SAVI, and NDVI indices leads to a decrease in the surface albedo. Areas with high

albedo indicate vegetation degradation and soil exposure. In the studied area, in regions classified as having high levels of degradation, the surface Albedo is high, while the vegetation cover index shows a lower value. These findings are consistent with those of studies conducted in other parts of the world [3].

The correlation analysis between the BSI and TGSI indices with albedo indicates a positive relationship. However, the intensity map of degradation obtained from these indices shows a significant difference from the field data, as confirmed by the Mann-Whitney U test with a significance level of 0.00. Therefore, these two indices cannot be used to monitor this area's degradation. But lamchin et al (2016), confirmed a strong correlation between albedo and TGSI indices in the evaluation of desertification of mongolin land with remote sensing technique [10].

The examination of the Root Mean Square Error (RMSE) between the models obtained from the Albedo index and each of the spectral indices derived from remote sensing and field data revealed that the intensity of degradation derived from the MSAVI-Albedo index has the highest proximity to the field data with an RMSE of 0.65. This indicates a closer agreement than the other indices, NDVI and SAVI, which is consistent with Zandler et al.'s study in 2022, which demonstrated that the NDVI index is not suitable for dryland with sparse vegetation covers [40]. Furthermore, the intensity of degradation obtained from the NDVI-TGSI model with an RMSE of 0.66 is higher than other models, indicating that the degradation intensity map derived from these two indices has higher accuracy compared to vegetation cover indices with Albedo [41].

The classification of degradation intensity in the Rutak Saravan area based on the Albedo-MSAVI model, which had less error with the field data, indicates that 0.46% of the area is not classified as degraded, 4.11% falls into the low degradation class, 13.93% is classified as moderate degradation, 41.78% is classified as severe degradation, and 39.72% of the area is classified as very severe degradation. The behavior of TGSI and BSI with NDVI was similar to that of the Albedo index. This indicates that TGSI and BSI indices can also be used in preparing the destruction intensity map. The results of comparing and testing the accuracy of the model prepared with albedo index, vegetation indices, TGSI, and BSI indices with NDVI index using field data show that the model prepared with TGSI index and NDVI has a high accuracy in evaluating the severity of destruction with an RMSE of 0.663. The study suggests that the model prepared based on the Normalized Difference Vegetation Index (NDVI) and the Top-soil Grain Size Index (TGSI) has a high accuracy for identifying the severity of destruction. This model was found to be more accurate after the model was prepared based on the MSA-VI-Albedo index. The NDVI-TGSI model had an RMSE of 0.663 with the field data. The results indicated that 0.25% of the area is not classified as degraded, 2.43% falls into the low degradation class, 10.11% is classified as moderate degradation, 39.21% is classified as severe degradation, and 48.01% of the area is classified as very severe degradation (Table 5).

These are inconsistent with studies conducted in the study area using other standard models to map desertification intensity, which identified the loss of vegetation cover as one of the main causes of degradation and desertification [19, 22].

The results of determining desertification intensity based on the IMDPA model in the study area [22] showed that one of the critical factors is the loss of vegetation cover. It can be concluded that the best spectral indices with the lowest RMSE for mapping degradation intensity, which also have the highest correlation with the Albedo index, are MSAVI, TGSI-NDVI, SAVI, and NDVI. Therefore, it can be said that the results of classifying the degradation intensity of the study area in the present research based on the Albedo model and each of the investigated indices, especially MSAVI, are in a favorable position for identifying degraded areas and determining degradation intensity classes. This finding is consistent with other studies in this field [12, 13, 35].

## 5. Conclusions

One of the objectives of studying various indices for determining the intensity of degradation in different areas is to achieve the highest accuracy in using the desired index. In this study, the relationship between different spectral indices and albedo was evaluated using remote sensing technology and linear regression.

This method effectively identifies breakpoints between classifications with the optimized JENKS algorithm based on the nature of the data and their inherent grouping. In the present study, the degradation intensity in the dry region of Balochistan, Iran, was extracted using remote sensing techniques and based on multispectral Sentinel-2 images. The degradation intensity was examined based on the spectral reflectance from the ground surface and the spatial resolution of 10 meters.

Considering the high spatial resolution of Sentinel-2 images in the present study and the appropriate identification of degraded areas for mapping degradation intensity, it is suggested, based on the results of this research, to use a combination of the Albedo model and other spectral indices for monitoring and mapping degradation intensity in other dry regions as well. Given that monitoring and controlling degraded and desertified areas and preparing degradation intensity maps based on conventional models are challenging, costly, and require extensive field activities, using fast and accurate models seems highly necessary.

Furthermore, since in most studies, geomorphological units are used as working units for mapping degradation and desertification intensity, it cannot be said that the degradation intensity map obtained from these models is sufficiently accurate at small scales due to the specific land characteristics, vegetation type, and soil moisture. The development and use of such a model in the future can be a novel step toward rapidly identifying and monitoring degraded areas in remote and inaccessible regions. Moreover, it will be effective in identifying sources of dust production and susceptibility to wind erosion. It is predicted that the application of Sentinel-2 MSI (S2) data will greatly improve the estimation of degradation intensity at a local and regional scales.

## Supporting information

**S1 Appendix. The process of conducting field studies to determine the intensity of degradation in each plot.**
(DOCX)

**S1 Graphical abstract.**
(PDF)

## Author Contributions

**Data curation:** Akhtar Ebrahimi.

**Investigation:** Farhad Zolfaghari.

**Methodology:** Marzieh Ghodsi.

**Resources:** Fatemeh Narmashiri.

**Software:** Farhad Zolfaghari.

**Supervision:** Farhad Zolfaghari.

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
