## [Decision Letter · Decision Letter 0]

28 Sep 2023

PONE-D-23-27523Assessing the accuracy of spectral indices obtained from Sentinel images using field research to estimate land degradationPLOS ONE

Dear Dr. zolfaghari, 

Thank you for submitting your manuscript to PLOS ONE. After careful consideration, we feel that it has merit but does not fully meet PLOS ONE’s publication criteria as it currently stands. Therefore, we invite you to submit a revised version of the manuscript that addresses the points raised during the review process. Please submit your revised manuscript by Nov 12 2023 11:59PM. If you will need more time than this to complete your revisions, please reply to this message or contact the journal office at plosone@plos.org. Please include the following items when submitting your revised manuscript:A rebuttal letter that responds to each point raised by the academic editor and reviewer(s). You should upload this letter as a separate file labeled 'Response to Reviewers'.A marked-up copy of your manuscript that highlights changes made to the original version. You should upload this as a separate file labeled 'Revised Manuscript with Track Changes'.An unmarked version of your revised paper without tracked changes. You should upload this as a separate file labeled 'Manuscript'.

We look forward to receiving your revised manuscript.

Kind regards,

Chun Liu

Academic Editor

PLOS ONE

Journal Requirements:

4. We note that Figures 1, 3 and 4 in your submission contain map/satellite images which may be copyrighted. All PLOS content is published under the Creative Commons Attribution License (CC BY 4.0), which means that the manuscript, images, and Supporting Information files will be freely available online, and any third party is permitted to access, download, copy, distribute, and use these materials in any way, even commercially, with proper attribution. For these reasons, we cannot publish previously copyrighted maps or satellite images created using proprietary data, such as Google software (Google Maps, Street View, and Earth). For more information, see our copyright guidelines: http://journals.plos.org/plosone/s/licenses-and-copyright.

a. You may seek permission from the original copyright holder of Figures 1, 3 and 4 to publish the content specifically under the CC BY 4.0 license.  

5. Please ensure that you refer to Figure 4 in your text as, if accepted, production will need this reference to link the reader to the figure.

6. We note you have included a table to which you do not refer in the text of your manuscript. Please ensure that you refer to Table 5 in your text; if accepted, production will need this reference to link the reader to the Table.

Reviewers' comments:

Reviewer's Responses to Questions

**Comments to the Author**

1. Is the manuscript technically sound, and do the data support the conclusions?

Reviewer #1: Partly

Reviewer #2: Partly

Reviewer #3: Partly

2. Has the statistical analysis been performed appropriately and rigorously? 

Reviewer #1: Yes

Reviewer #2: No

Reviewer #3: I Don't Know

3. Have the authors made all data underlying the findings in their manuscript fully available?

Reviewer #1: Yes

Reviewer #2: Yes

Reviewer #3: No

4. Is the manuscript presented in an intelligible fashion and written in standard English?

Reviewer #1: Yes

Reviewer #2: Yes

Reviewer #3: Yes

5. Review Comments to the Author

Reviewer #1: The manuscript is well written and described the need of monitoring and classification of land degradation and its occurrence using Sentinel images. However, as stated in the title, .........field research to estimate land degradation. I have not found any accuracy assessment using the field data.

1. References cited in the text are not as per PLOSONE Journal format.

2. Expand the abbreviations of indices at the time of first occurrence.

3. Line 96, correct the word sentinel as sentinel2.

4. Several studies addressed the issue of land degradation and desertification with the use of Albedo, NDVI and TGSI (Ma et al., 2011; Chen et al., 2021; Bazgeer et al., 2019; Aramesh et al., 2022; Lamqadem et al., 2018) using sentine2 and other satellite datasets. Describe the goals and objectives of the study with innovation and how it differs with other studies.

5. The authors described in the Line, 182, that the accuracy of the study was tested with 100 ground control points and field observations. However, the details on ground data collection, parameters used, use of any portable or suitable devices for the measurement of albedo, and other tested vegetation indexes are not clear.

Reviewer #2: This manuscript does not contain sufficient novelty to be published in Plos One. There are several studies on land degradation. Since this paper does not investigate the recent literature (2021,2022 and 2023) comprehensively, it does not convincingly address a literature gap.

The introduction section is weak in terms of reflecting what is already available in the literature with a land degradation focus; therefore, it is not clear if authors are filling a literature gap or not.

The paper fails to provide state-of-the-art methods and analysis with these methods as a benchmark, which is an important weakness.

They do not provide clear scientific questions.

The discussion section is very superficially written, as a major outcome of not including the recent papers and comprehensively analyzing them.

It is not clear why the selected study area should be interest of global readers.

Reviewer #3: The manuscript deals with an important subject but explanation of some points is needed.

All acronyms must be defined/described the first time they appear. Many vegetation and soil indices were used and were not defined. The proper references to such indices are also needed. Additionally, their main usage must be addressed, for instance, BSI (Bare Soil Index) is a measure of bare soil in a region and is derived by…

When describing Sentinel 2 data, please state the bands (and the region of the spectrum they cover).

Line 165 presents a brief description of the study area. This section could be improved, by adding more information about the vegetation (are annual and perennial species grassland pecies?).

Describe Equation 7 including “a” coefficient.

As it appears on the title of the manuscript, the work done in the field, at the study area, have to be described. How did you organize the data you collected in the field? Did you generate a reference map?

Please explain why you did check correlation between BSI and TGSI with NDVI (line 220). The text shows that you assume that albedo is the information used as an indicator of the absence of vegetation. What changes when you use the relations BSIxNDVI and TGSIxNDVI? Please refer to that in the discussion section.

You did not comment Table 5.

Line 295: You stated the higher accuracy (less error for consistency with field survey) was with MSAVI-albedo. Please explain why next you stated that “the NDVI-TGSI model (RMSE=0.66) outperformed other models in terms of accuracy”.

Please explain what IMDPA model is.

Discussion section could show more references to similar works in other regions of the Earth.

6. PLOS authors have the option to publish the peer review history of their article (what does this mean?). If published, this will include your full peer review and any attached files.

Reviewer #1: No

Reviewer #2: No

Reviewer #3: No

---

## [Author Response · Author response to Decision Letter 0]

11 Jan 2024

Amendment Note about PONE-D-23-27523

" Assessing the accuracy of spectral indices obtained from Sentinel images using field research to estimate land degradation "

To Editor

Dear respected editor and reviewers,

 The authors have made the necessary changes based on the reviewers’ comments and provided a point-by-point response to address the comments and revisions needed. The supporting information file “S1 Appendix” was added to answer a question posed by a referee regarding the field study. The authors note that all relevant data are within the manuscript and its supporting information files. If there are any further queries, they have requested that they be informed.

Best Regards,

Farhad Zolfaghari

(On behalf of all authors)

To academic editor

1- Manuscript style requirements were checked.

2- The implementation of the field project did not require a permit due to the region’s governing laws, and public access to the area is free.

3- the statement “The funders had no role in study design, data collection and analysis, decision to publish, or preparation of the manuscript. The authors received no specific funding for this work“, were added to “Declaration of interest“ section.

4-You have raised an important point in relation with images which may be copyrighted. Figure 1 in the manuscript was downloaded from NASA Earth Observatory (public domain):http://earthobservatory.nasa.gov/ (public domain).

The sentinel data required for figures 3 and 4 was obtained from the Open Access Hub of the European Space Agency, which is publicly accessible at https://scihub.copernicus.eu/dhus/#/home 1. After extracting the analyzed indicators in Snap and ArcGIS software, figures 3 and 4 were produced, which represent the output of the software

5- Many thanks for pointing this out. I have revised it in text.

6- Many thanks for pointing this out. I have revised it in text.

7- Many thanks for pointing this out. I have revised it in text.

To Reviewer #1

1. References cited in the text are not as per PLOSONE Journal format.

Reply: Agreed, as I have revised it in text.

2. Expand the abbreviations of indices at the time of first occurrence.

Reply: Many thanks for pointing this out. I have revised it in text. 

3. Line 96, correct the word sentinel as sentinel2.

Reply: Thank you for your thorough review and salient observation. I have revised it in text. 

4. Several studies addressed the issue of land degradation and desertification with the use of Albedo, NDVI and TGSI (Ma et al., 2011; Chen et al., 2021; Bazgeer et al., 2019; Aramesh et al., 2022; Lamqadem et al., 2018) using sentine2 and other satellite datasets. Describe the goals and objectives of the study with innovation and how it differs with other studies.

Reply: Many thanks for pointing this out. Validation of different spectral indices and field studies to determine the best spectral index for accurately assessing land degradation. This validation is essential due to the characteristics of the type of cover and the unique conditions of areas affected by the destruction. this process requires a scientific approach to identify the factors that contribute to land degradation and determine their intensity. The validation process involves a rigorous evaluation of the factors that contribute to degradation, including soil, vegetation, wind erosion, and climate criteria. this validation process is crucial because it helps to ensure that the best spectral index is used to accurately assess land degradation. The explanation has been added to the introduction section, which provides an overview of the research topic and highlights its significance. (Pages: 4 and 5, Lines: 98-109)

5. The authors described in the Line, 182, that the accuracy of the study was tested with 100 ground control points and field observations. However, the details on ground data collection, parameters used, use of any portable or suitable devices for the measurement of albedo, and other tested vegetation indexes are not clear.

Reply: Thank you for your thorough review and salient observation. for example, for plat 1, the climate indicator was calculated based on the geometric mean of the weights of three indices including precipitation, drought index, and drought event: 

 Moreover, the degradation severity of the plat 1 was calculated using the geometric mean weight of four relevant indicators (climate, soil, wind erosion, and vegetation):

Then, according to the table below, the degradation intensity of each plot was determined.

Class of desertification Range of score

Low 1-1.24

Medium 1.25-1.49

High 1.5-1.74

Very High 1.75-2

The full description of the field study is added in Supporting Information files as “S1 Apendix”

To Reviewer #2

- This manuscript does not contain sufficient novelty . . .

Reply: The reviewer raises an interesting concern. However, we feel the reviewer may not fully comprehend the scope of the work. This study is designed to address the question of how different spectral indices extracted from satellite images can be utilized to identify areas affected by destruction. It is important to note that the applicability of these indices may vary across different geographical regions. To achieve this objective, the relationship between the albedo index and indicators associated with bare soil surface and vegetation was examined to quantify the extent of destruction. Additionally, a field-based approach was employed, involving the investigation of 100 plots, to determine which indicators yield the most accurate results in assessing the intensity of destruction. These plots were used to evaluate criteria related to soil, vegetation, wind erosion, and climate within the study area. Finally, the Mann-Whitney statistical model was employed to identify and introduce the best model derived from spectral indices

To Reviewer #3

1. All acronyms must be defined/described the first time they appear.

Reply: Many thanks for pointing this out. I have revised it in text.

2. Many vegetation and soil indices were used and were not defined. The proper references to such indices are also needed. Additionally, their main usage must be addressed, for instance, BSI (Bare Soil Index) is a measure of bare soil in a region and is derived by…

Reply: Thank you for bringing this to our attention. We have added the explanation and Table 2 to the methodology section, as per your suggestion. The relevant pages are 7-9, and the lines are 169-188.

3. When describing Sentinel 2 data, please state the bands (and the region of the spectrum they cover).

Reply: Thank you for your thorough review and salient observation. The explanation and Table 1 have been added to the document (Page: 7, Lines: 155-167).

4. Line 165 presents a brief description of the study area. This section could be improved, by adding more information about the vegetation (are annual and perennial species grassland pecies?).

Reply: Thank you for your constructive comments and suggestions. The vegetation in the area is predominantly comprised of bush and shrub species, which are sparsely distributed in the flood plains. Due to inadequate precipitation, several areas of the vegetation cover have desiccated. Herbaceous and annual species have nearly vanished, leading the land that was previously utilized as pasture for light livestock to transform into completely arid regions with scanty bush cover. Table 3 presents the floristic list of the study area. The materials and methods section has been updated with this information. (Page: 6, Lines: 130-134). 

Floristic list of the study area

ردیف نام علمی گونه ردیف نام علمی گونه 

1 Tamarix aphylla 11 Anabasis steifira

2 Hamada salicornica 12 Atriplex leucoclad 

3 Haloxylon persicum 13 Cusinia gadrosisa

4 Caparis spinosa 14 Echinops sedrosiaca

5 Aellenia ariculata 15 Allysum dasycarpum

6 Aellenia subaph 16 Citrullus colocynthis

7 Agriophyllum minus 17 Carex physodes

8 Peganum harmala 18 Bromus tectorum

9 Cynodon dactylon 19 Calligonum bungei

10 Cousinia stocksii 20 Nitraria schoberi

5. Describe Equation 7 including “a” coefficient.

Reply: Many thanks for pointing this out. The (a) value is the slope of the orthogonal lines found in the NDVI, SAVI, MSAVI, BSI, and TGSI separately with Albedo relationship, as added in the text.

6. As it appears on the title of the manuscript, the work done in the field, at the study area, have to be described. How did you organize the data you collected in the field? Did you generate a reference map?

Reply: Thank you for your constructive comments. In our field operations, we randomly selected 100 plots of 10 x 10 square meters in different parts of the study area. For each plot, we evaluated four criteria: vegetation cover, wind erosion, climate, and soil quality. We used indicators defined for each criterion based on an Iranian Model of Desertification Potential Assessment (IMDPA) model. The summary of the field model we investigated is presented in S1 Appendix due to the article’s increased volume.

7. Please explain why you did check correlation between BSI and TGSI with NDVI (line 220).

Reply: Thank you for your constructive feedback. We believe that it would be beneficial to examine and investigate the correlation between various indices to identify the most effective spectral indices that can indicate the severity of degradation. As stated in the introduction, this research aims to address the question of identifying the optimal spectral index for determining degradation intensity. During our evaluation of various indicators with field data, we were intrigued by how we could assess the results of these indicators with field data. As revised in the text (page: 15, lines 335-347)

8. The text shows that you assume that albedo is the information used as an indicator of the absence of vegetation. What changes when you use the relations BSIxNDVI and TGSIxNDVI? Please refer to that in the discussion section.

Reply: The reviewer raises an interesting concern. I have revised it and have added to the text (Page: 19, Line: 428-440).

9. You did not comment Table 5.

Reply: Agree. Many thanks for pointing this out. I have revised it in text.

10. Line 295: You stated the higher accuracy (less error for consistency with field survey) was with MSAVI-albedo. Please explain why next you stated that “the NDVI-TGSI model (RMSE=0.66) outperformed other models in terms of accuracy”.

Reply: Many thanks for pointing this out. I have revised it (Page: 15, Lines: 335-347). We think these changes are now better. We hope that you agree.

11. Please explain what IMDPA model is.

Reply: The explanation of the Iranian Model Desertification Potential Assessment (IMDPA) has been added to the S1 Appendix supporting file. 

12. Discussion section could show more references to similar works in other regions of the Earth.

Reply: Many thanks for pointing this out. as I have revised it in text (Page: 18, Lines: 407-415).

---

## [Decision Letter · Decision Letter 1]

9 May 2024

PONE-D-23-27523R1Assessing the accuracy of spectral indices obtained from Sentinel images using field research to estimate land degradationPLOS ONE

Dear Dr. zolfaghari,

Thank you for submitting your manuscript to PLOS ONE. After careful consideration, we feel that it has merit but does not fully meet PLOS ONE’s publication criteria as it currently stands. Therefore, we invite you to submit a revised version of the manuscript that addresses the points raised during the review process.

We look forward to receiving your revised manuscript.

Kind regards,

Yangyang Xu

Academic Editor

PLOS ONE

Journal Requirements:

Reviewers' comments:

Reviewer's Responses to Questions

**Comments to the Author**

1. If the authors have adequately addressed your comments raised in a previous round of review and you feel that this manuscript is now acceptable for publication, you may indicate that here to bypass the “Comments to the Author” section, enter your conflict of interest statement in the “Confidential to Editor” section, and submit your "Accept" recommendation.

Reviewer #1: All comments have been addressed

2. Is the manuscript technically sound, and do the data support the conclusions?

Reviewer #1: Yes

3. Has the statistical analysis been performed appropriately and rigorously? 

Reviewer #1: N/A

4. Have the authors made all data underlying the findings in their manuscript fully available?

Reviewer #1: Yes

5. Is the manuscript presented in an intelligible fashion and written in standard English?

Reviewer #1: (No Response)

6. Review Comments to the Author

Reviewer #1: Authors attempted to enhance the quality of the manuscript. However, minor modifications as follows: 1. Line 99: Expand the abbrivation "IMDPA", as it is first occurrence, in the manuscript. Provide a reference.

2. Table 1: spectral bands of MSI sensor - As authors used onlt 10 m bands of sentinel image. This information is available at sentinel-2 website, hence the reference is enough. It is recommended to remove.

3. Table 5: Provide scale of values to seviriety degradation class. (for example: low ??, describe the range in scale)

4. The description provided in Appendix-S1 is a key for field data and its analysis. It should be in the methods section of the main/manuscript. However, the tables can be provided in S1 Appendex.

7. PLOS authors have the option to publish the peer review history of their article (what does this mean?). If published, this will include your full peer review and any attached files.

Reviewer #1: No

---

## [Author Response · Author response to Decision Letter 1]

10 May 2024

Amendment Note about PONE-D-23-27523

" Assessing the accuracy of spectral indices obtained from Sentinel images using field research to estimate land degradation "

To Editor

Dear respected editor and reviewers,

 The authors have made the necessary changes based on the reviewers’ comments and provided a point-by-point response to address the comments and revisions needed. An additional reference has been incorporated into the Methods section (page 9, line 196). This new source is now listed as reference number 39 in the references section (page 26, line 571). 

The authors note that all relevant data are within the manuscript and its supporting information files. If there are any further queries, they have requested that they be informed.

Best Regards,

Farhad Zolfaghari

(On behalf of all authors)

To Reviewer #1

1. Authors attempted to enhance the quality of the manuscript. However, minor modifications as follows: 1. Line 99: Expand the abbreviation "IMDPA", as it is first occurrence, in the manuscript. Provide a reference.

Reply: Agreed, as I have revised it in text. (Page: 4, Lines: 99-100)

2. Table 1: spectral bands of MSI sensor - As authors used onlt 10 m bands of sentinel image. This information is available at sentinel-2 website; hence the reference is enough. It is recommended to remove.

Reply: Many thanks for pointing this out. I have revised it in text. (Page: 6, Line: 146)

3. Table 5: Provide scale of values to severity degradation class. (for example: low ??, describe the range in scale)

Reply: Thank you for your thorough review and salient observation. I have revised it in text. (Page: 13, Line: 300).

4. The description provided in Appendix-S1 is a key for field data and its analysis. It should be in the methods section of the main/manuscript. However, the tables can be provided in S1 Appendix.

Reply: Thank you for your thorough review and salient observation. The description of the field study is added in methodology section. (Pages: 9 and 10, Lines: 199-219)

---

## [Decision Letter · Decision Letter 2]

5 Jun 2024

Assessing the accuracy of spectral indices obtained from Sentinel images using field research to estimate land degradation

PONE-D-23-27523R2

Dear Dr. zolfaghari,

We’re pleased to inform you that your manuscript has been judged scientifically suitable for publication and will be formally accepted for publication once it meets all outstanding technical requirements.

Kind regards,

Yangyang Xu

Academic Editor

PLOS ONE

Additional Editor Comments (optional):

Reviewers' comments:

Reviewer's Responses to Questions

**Comments to the Author**

1. If the authors have adequately addressed your comments raised in a previous round of review and you feel that this manuscript is now acceptable for publication, you may indicate that here to bypass the “Comments to the Author” section, enter your conflict of interest statement in the “Confidential to Editor” section, and submit your "Accept" recommendation.

Reviewer #1: All comments have been addressed

2. Is the manuscript technically sound, and do the data support the conclusions?

Reviewer #1: Yes

3. Has the statistical analysis been performed appropriately and rigorously? 

Reviewer #1: (No Response)

4. Have the authors made all data underlying the findings in their manuscript fully available?

Reviewer #1: (No Response)

5. Is the manuscript presented in an intelligible fashion and written in standard English?

Reviewer #1: (No Response)

6. Review Comments to the Author

Reviewer #1: Authors improved the manuscript - Assessing the accuracy of spectral indices obtained from Sentinel images using field research to estimate land degradation. It is recommended for publication.

7. PLOS authors have the option to publish the peer review history of their article (what does this mean?). If published, this will include your full peer review and any attached files.

Reviewer #1: No

---

## [Editor Report · Acceptance letter]

20 Jun 2024

PONE-D-23-27523R2 

PLOS ONE

Dear Dr. zolfaghari, 

I'm pleased to inform you that your manuscript has been deemed suitable for publication in PLOS ONE. Congratulations! Your manuscript is now being handed over to our production team.

Kind regards, 

on behalf of

Dr. Yangyang Xu 

Academic Editor

PLOS ONE